# Risk Prediction and Variable Analysis of Pine Wilt Disease by a Maximum Entropy Model



**Zhuoqing Hao** [1,2], **Guofei Fang** [3,4,*], **Wenjiang Huang** [1,2,5], **Huichun Ye** [1,5,*], **Biyao Zhang** [1] **and Xiaodong Li** [3,4]

1 State Key Laboratory of Remote Sensing Science, Aerospace Information Research Institute, Chinese Academy of Sciences, Beijing 100094, China; haozhuoqing20@mails.ucas.ac.cn (Z.H.); huangwj@radi.ac.cn (W.H.); zhangby@aircas.ac.cn (B.Z.)
2 University of Chinese Academy of Sciences, Beijing 100049, China
3 Center for Biological Disaster Prevention and Control, National Forestry and Grassland Administration, Shenyang 110034, China; lxdong1221@126.com
4 Key Laboratory of National Forestry and Grassland Administration on Forest and Grassland Pest Monitoring and Warning, Shenyang 110034, China
5 Key Laboratory of Earth Observation of Hainan Province, Hainan Research Institute, Aerospace Information Research Institute, Chinese Academy of Sciences, Sanya 572029, China
* Correspondence: fgfly@163.com (G.F.); yehc@aircas.ac.cn (H.Y.)

**Abstract:** Pine wilt disease (PWD) has caused a huge damage to pine forests. PWD is mainly transmitted by jumping diffusion, affected by insect vectors and human activities. Since the results of climate change, pine wood nematode (PWN—*Bursaphelenchus xylophilus*) has begun invading the temperate zones and higher elevation area. In this situation, predicting the distribution of PWD is an important part of the prevention and control of the epidemic situation. The research established the Maxent model to conduct a multi-angle, fine-scale prediction on the risk distribution of PWD. We adjusted two parameters, regularization multiplier (RM) and feature combination (FC), to optimize the model. Influence factors were selected and divided into natural, landscape, and human variables, according to the physical characteristics and spread rules of PWD. The middle-suitability regions and high-suitability regions are distributed in a Y-shape, and divided the study area into three parts. The high-suitability areas are concentrated in the region with high temperature, low elevation, and intensive precipitation. Among the selected variables, natural factors still play the most important role in the distribution of the disease, and human factors and landscape factors are also worked well. The permutation importance of factors is different due to differences in climate and other conditions in different regions. The multi-angle, fine-scale model can help provide useful information for effective control and tactical management of PWD.

**Keywords:** pine wilt disease; maximum entropy model; influencing factors; suitable probability

## 1. Introduction

Pines are coniferous trees of the *Pinus* genus with high economic value, playing an important role in soil and water conservation [1,2]. Pine wilt disease (PWD) is caused by the pine wood nematode (PWN—*Bursaphelenchus xylophilus*) and is a worldwide forest disease. As one of the most important forest pests in the quarantine list in more than 40 countries [3], PWD is a serious threat to pine forests globally [4]. When nematodes settle in the host trees, the pine trees die in less than three months [5,6]. PWD has the characteristics of easy transmission, rapid onset, a high fatality rate, and difficulty in prevention and control. Masson pine (*Pinus massoniana*) is a susceptible species in China [6]. The invasion of PWD has damaged the Chinese forest ecosystem, causing tree species conversion, wildlife habitat destruction, soil and water conservation, and biodiversity loss [7]. In addition, it also seriously affects Chinese import and export trade [8]. Biodiversity degradation in

forest ecosystems as a result of non-native pathogens has received a lot of attention from forest managers. PWD has become one of the most serious forest diseases in the world [9]. Sustainable forest management and effective pest control of pine forests requires rigorous exploration of potential PWD risk areas.

PWN supposedly originated in North America, and was then brought to Asia through the interregional wood trade in the early twentieth century [10]. PWN invaded Japan in 1905, China in 1982, and Korea in 1988. After causing widespread death in Asia, PWD first appeared in Europe in 1999 [11]. In 1982, PWD was first discovered in China on black pine (*Pinus thunbergii Parl*) of Nanjing, Jiangsu Province [12]. PWD expanded to the surrounding counties at a rate of 7000~10,000 hm$^2$ per year in China [13]. Within a few years, the disaster spread from Nanjing to Jiangsu, Zhejiang, Anhui, and other places. PWD spread rapidly in many districts, causing a serious damage in the pine forest. By 1999, the number of diseased and dead trees reached 5.5 million [2]. According to PWD surveys in the spring and autumn of 2018, there are 18 provinces and 589 counties which have been invaded by PWD, covering an area of 650,000 hectares [14,15]. The spread of PWD was characterized by wide range and long distance, causing huge losses to China's ecological environment, natural landscape, and social economy.

PWD was mainly transmitted by jumping diffusion [16], affected by insect vectors and human activities. The short-distance dispersal of PWD is affected by temperature and drought, and human-mediated dispersal promotes the expansion of PWD distribution, via long-distance dispersal [17]. Studies have shown that areas with an average annual temperature of 10–14 °C, especially the southern parts of Yellow River in China, are especially susceptible [2]. The medium- and high-suitability areas for PWN are mainly distributed in the low-altitude area, below 700 m. However, the spread of PWD has broken through the traditional theory of suitable limits in the past few years. The globalization of trade and climate change are increasing the opportunity for further incursion and expansion of PWN around the world [18]. Climate change is influencing vegetation's type and distribution globally [19], since the development of temperature and drought. The increases in mean temperature and drought affect the natural ecosystem and could promote the spread of PWD into areas where the risk is low under current climatic conditions [20]. By 2030, it has been predicted that PWD could spread across 8–34% of Europe, with the cumulative value of lost forestry stock estimated at EUR 22 billion if PWN is not controlled [21]. At present, it has begun invading the temperate zones (e.g., Liaoning province) and high-elevation areas (e.g., Qinling Mountains) in China.

To prevent the pandemic spread of focal pathogens, it is crucial to identify potential risk areas and prevalence status. The studies mentioned above are mostly based on climate variables in large scale, which have revealed the distribution of PWD under climate change scenarios. Many papers studied a certain factor and explored the relationship between the factor and the spread of PWD. For instance, human-induced dispersal plays a fundamental role in the spread of the PWD, and transportation of trade (i.e., maritime transportation, roads, and railways) is an important pathway for the nematode [17,22]. Landscape pattern has a significant effect on the dispersal of the disaster [23–25]. There are many factors affected the spread of PWD, and the effect of different factors is related to the research scale. However, the current literature lacks multi-angle analysis—including a consideration of the characteristics of climate and trade in study areas. Moreover, the present studies are mainly large-scale, macro predictions, on international, intercontinental, and national scales [2,26,27]. The regional characteristics affect the reliability of the prediction results a great extent because the development of PWD has regional characteristics. Additionally, current research also shows a lack of analysis of factor response changes in different environments [23]. All of these problems influence the accuracy of prediction results.

In order to solve these problems, this study conducted a multi-angle, fine-scale study on the risk distribution of PWD. We carried out prediction and analysis with regional characteristics in the municipal scale, considering climate, topography, landscape pattern, traffic, and other factors. The aim of this study to explore the distribution characteristics of

PWN at small scale. We used the Maxent model to predict the suitable probability of PWN, and then analyzed the distribution characteristics and the factor response. This research can help understand the underlying pathological mechanisms of PWN, and can provide useful information for strategic and tactical management.

## 2. Materials and Methods

### 2.1. Study Area

The study area is located in Yichang City, Hubei Province (Figure 1a,b). The study area has a typical subtropical monsoon climate, with rainy and hot weather occurring within the same season. Both high temperature and precipitation are concentrated in July. The annual average rainfall is more than 1000 mm, and the annual average temperature is 14.1~17.1 °C. The climate is warm and humid, with four distinct seasons, characterized by transition from north to south. The forest coverage rate in study area reaches 65.7%. The forest trees are mainly pines, including oak and citrus trees. The pines are mainly Masson pine (*Pinus massoniana* Lamb) and slash pine (*Pinus elliottii*). The Masson pine is the natural host of PWN and Japanese pine sawyer (*Monochamus alternatus*). Japanese pine sawyer is densely distributed in Yichang City. It has a superimposing effect on the spread of PWD. The high temperature and rainfall provide suitable climatic conditions for the breeding period of PWN. In addition, the study area has many landform types, forming a landform pattern of "six hills, three flats, and one mountain". The terrain is high in the west and low in the east. The altitude is generally below 300 m. Dead trees appeared in Yiling of Yichang City in 2000, and they were confirmed as PWD in 2006 [28]. In the past 10 years, PWD has become a major biological disaster in this region, causing irreversible damage to the ecological and landscape environments.

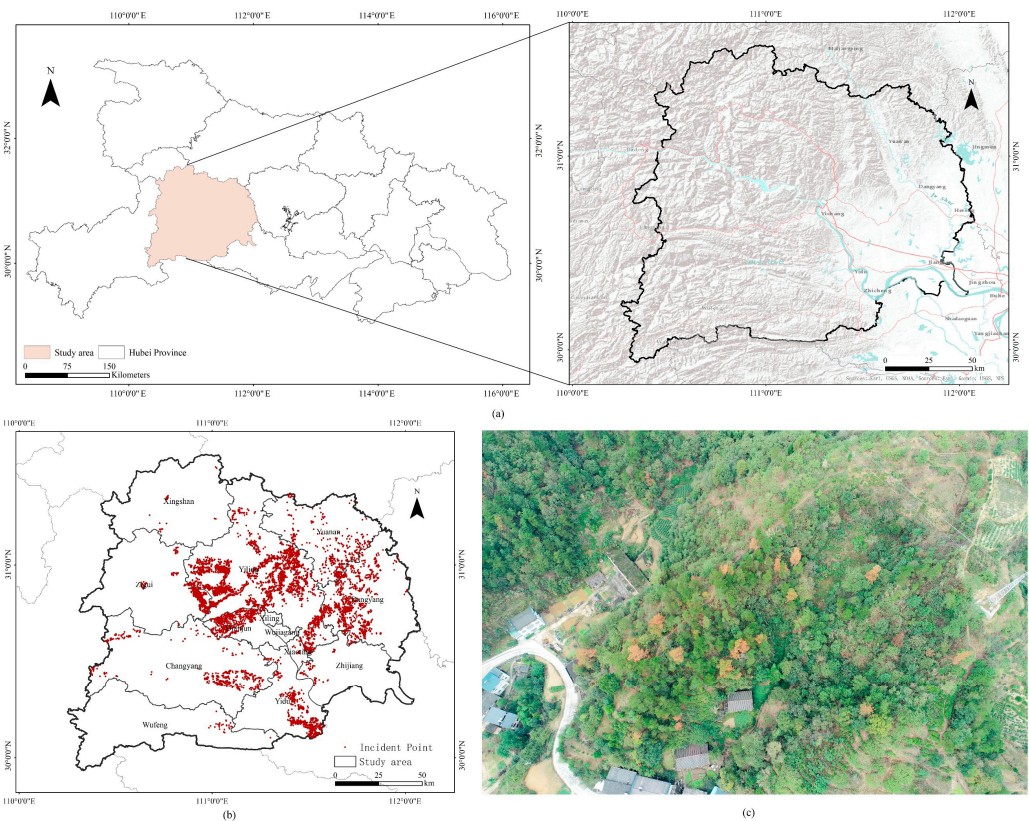

**Figure 1.** Overview of the study area. (**a**) The study area located in Yichang City, Hubei Province. (**b**) Distribution of occurrence point of PWD. (**c**) Field photo of PWD.

## 2.2. Data

Species occurrence data are the basis of potential suitable distribution predictions. The accuracy and reliability of the occurrence data are directly related to the plausibility of the predicted results. The occurrence points PWD in 2019 were provided by the Forestry and Grassland Diseases and Pests Control Station, from the National Forestry and Grassland Administration (NFGA) of the People's Republic of China. They were collected by the forestry bureaus of various districts and counties in China. Figure 1c shows the location of the occurrence data which we gathered in this study. We divided the occurrence points according to 7:3, as train data and test data, respectively.

Influence factors were selected and divided into natural, landscape, and human variables, according to the physical characteristics and spread rules of PWD. Among them, the precipitation and temperature are the primary natural factors affecting the development of PWD. Natural variables include average temperature, solar radiation, total precipitation, soil moisture, maximum temperature, minimum temperature, elevation, and landform. They are from multiple datasets and were collected by Google Earth Engine (GEE). Additionally, we calculated 6 landscape indexes, based on the type of landscape, by the FRAGSTATS software. The type of landscape data is that of FROM-GLC. The landscape indexes include the aggregation index (hereafter, AI), cohesion, division, patch density (hereafter, PD), Shannon's diversity index (hereafter, SHDI), and shape. The AI and cohesion are class metric measures that aggregate properties of the patches belonging to a single class in the landscape. We also calculated the distance to the Yangtze River and the distance to the road network, considering that human activities affect the spread of PWD. These two variables are human variables. The table of the data set is shown in Table 1.

**Table 1.** Influence factors used to predict the distribution of pine wilt diseasein Yichang City.

| Raster Dataset | | |
|---|---|---|
| **Variable** | **Spatial Resolution** | **Unit** |
| Average Temperature of July | 0.1 arc degrees | °C |
| Solar Radiation of June | 0.1 arc degrees | J/m$^2$ |
| Total Precipitation of July | 2.5 arc minutes | mm |
| Soil Moisture of July | 2.5 arc minutes | mm |
| Maximum Temperature of July | 2.5 arc minutes | °C |
| Minimum Temperature of July | 2.5 arc minutes | °C |
| Elevation | 30 m | m |
| Landform | 90 m | Class |
| Age of Stand | 30 m | Class |
| AI | 30 m | % (0, 100] |
| Cohesion | 30 m | % (0, 100] |
| Division | 30 m | PR $\geq$ 1 |
| PD | 30 m | n/100 ha |
| SHDI | 30 m | none |
| Shape | 30 m | none |
| Vector Calculation Result Data | | |
| **Variable** | **Vector Dataset** | **Unit** |
| Distance to Road | Road network shapefile | m |
| Distance to Water | Yangtze River shapefile | m |

## 2.3. Maxent Model and Evaluation

The Maxent model was constructed by S. J. Phillips in 2004 [29], and is a niche modelling approach based on the maximum entropy theory. This model was designed to find the probability distribution of the maximum entropy to estimate the probability distribution of the PWD using the known occurrence locations and the characteristics of variables. The model had the best performance and was the most widely used in species distribution models. Currently, the Maxent model has been used to analyze the prediction

of various biological fitness areas. This model has also been used in large-scale prediction of PWD. The research used SDM toolbox for ArcGIS and Maxent 3.4.1 to establish the Maxent model. PWD occurrence data were taken as sampling points, and 17 influence factors were taken as environmental data to analyze the disaster distribution and influence factors in Yichang city.

Model prediction is influenced by two parameters: regularization multiplier (RM) and feature combination (FC). The RM has an impact on the concentration of the output distribution. A smaller RM will result in a more localized output distribution, and a larger RM, plus the application range of prediction. FC constrains the calculated probability distribution. The default RM value is 1. In this study, we used RM constants of 0.5–5 with steps of 0.5. The feature types of this model include linear (L), quadratic (Q), product (P), threshold (T), and hinge (H). FCs combinations were allowed in model calibration, which has the potential to create complex fitted functions. Two feature types and three FC combinations—linear only (L); linear and quadratic (LQ); linear, quadratic, and product (LQH); hinge only (H); and linear, quadratic, hinge, product, and threshold features (LQHQT)—are used in this study. The performance of 50 Maxent models was compared, and we selected the optimal parameters. Each model used the average results of 10 repeated calculations to ensure the stability of the model.

The receiver operating characteristic curve (ROC) was generally used to evaluate the Maxent model's reliability in research. The area under the ROC curve (AUC) is used to evaluate the performance of the model. The range of AUC value is 0.5~1, which is positively correlated with the model's accuracy. Studies have shown that the higher the AUC value, the higher model's reliability. To ensure that the model has a low omission rate and high AUC, this study used the maximum total AUC and the prediction rate $(1 - \mathrm{OER})$ for accuracy verification, and used $(1 - \mathrm{OER}) + \mathrm{AUC}$ as the standard for evaluating the model. If a few models have the same $(1 - \mathrm{OER}) + \mathrm{AUC}$, then the model with the lowest complexity of FC is selected. The artificial classification method was adopted to reclassify the areas based on the existence probability of pine species and the PWD. The areas of habitat suitability were divided as follows: high suitability (0.6–1), medium suitability (0.4–0.6), low suitability (0.2–0.4), none (0–0.2).

*2.4. Landscape Metrics*

Landscape pattern plays an important role in the distribution of PWD. In this study, FROM-GLC land use type data and dominant tree species data of forest subclasses in Yichang city were used. Since the degree of species diversity and landscape fragmentation in an area affects the biological invasion of the area, we used six landscape metrics—AI, cohesion, division, PD, SHDI, and shape. These landscape metrics were calculated by the software FRAGSTATS—the Spatial Pattern Analysis Program for Categorical Maps (Version 4.2, Amherst, MA, USA). Additionally, the six metrics were calculated on the landscape scale. SHDI index could reflect the heterogeneity of the landscape, while SHDI increases, indicating that the types of patches increase or that the types of patches are evenly distributed in the landscape. Division is used to describe the degree of separation. PD, AI, and cohesion are "Aggregation metrics", and are used to describe the degree of aggregation. The more aggregated the landscape, the smaller the PD, and the greater the AI. Cohesion characterizes the connectedness of the landscapes. SHAPE reflects the complexity of the landscape, it increases as the shapes of the patches become more complex.

### 3. Results

*3.1. Model Optimization and Evaluation*

We performed correlation tests on variables and ranked them permutation importance, which reflected the importance of that factor to Maxent models. Variables with absolute correlation coefficient greater than 0.7 were eliminated to make the research variables more representative and the model more explanatory. According to the correlational analyses,

the seven variables—AI, SHDI, PD, division, shape, average temperature, and minimum temperature—were eliminated.

Natural factors and landscape factors, respectively, form two highly correlated variable groups, and two variable clusters with positive and negative correlations are formed in the two groups, respectively (Figure 2a). The natural variables formed two variable clusters: (1) soil moisture, precipitation, and elevation; (2) average temperature, maximum temperature, solar radiation, and minimum temperature. The two clusters showed a strong negative correlation. According to the importance ranking result (Figure 2b), elevation, precipitation, and maximum temperature all have high importance. Considering that the core influencing variables of PWD are temperature and precipitation, the two variables of minimum temperature and average temperature are replaced by maximum temperature. In addition, the 6 landscape factors are highly correlated. There were also two clusters: (1) AI and cohesion; (2) PD, SHDI, shape, and division. Obviously, cohesion has the highest importance. The correlation between AI and cohesion reached 0.896, the cohesion and the other landscape indexes without shape were negatively correlated, and the absolute correlation coefficient was greater than 0.7. The correlation between shape and other landscape indexes is low.

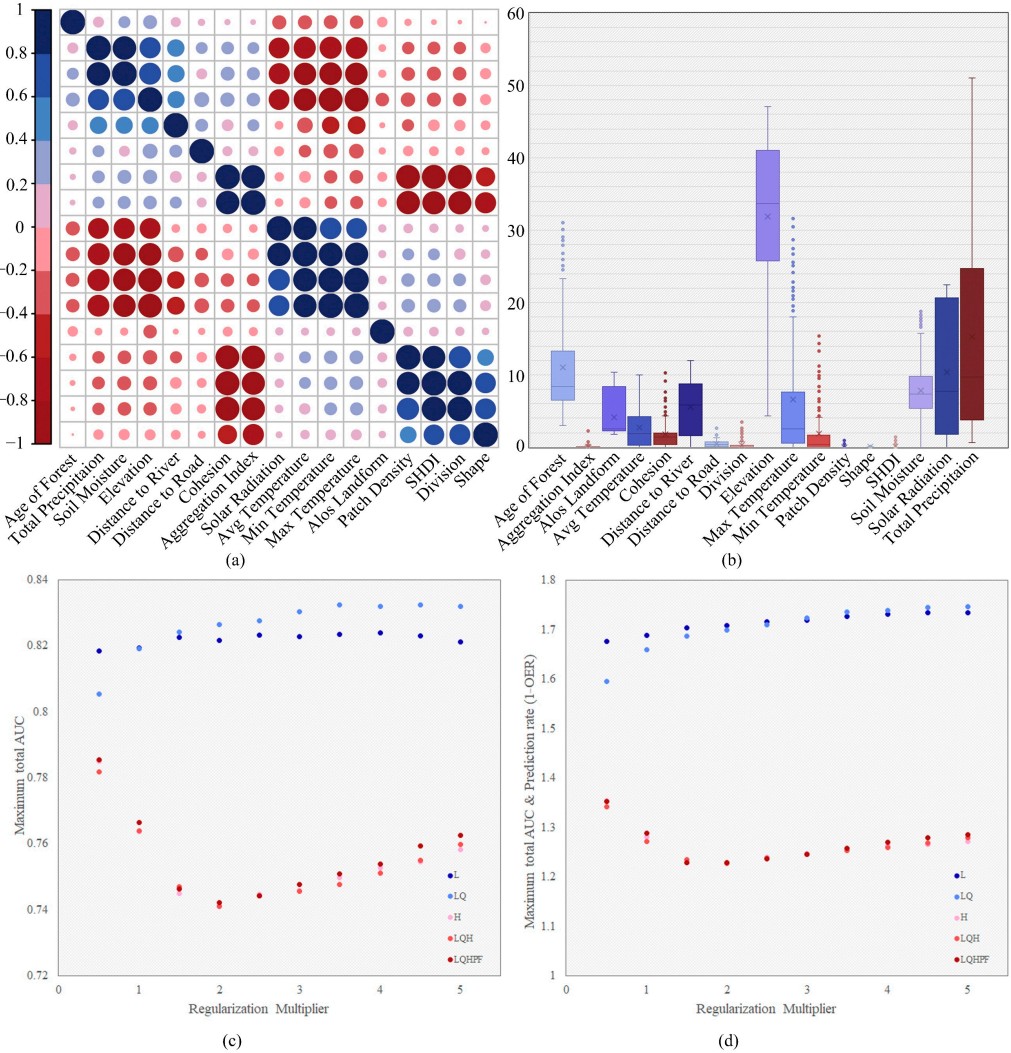

**Figure 2.** The results of correlation analysis and model evaluation of the 50 Maxent models run using different FC and RM. (**a**) Correlation coefficient of 17 influence factors; (**b**) average permutation importance of 50 Maxent models of influence factors; (**c**) maximum total AUC of 50 Maxent models; (**d**) maximum total AUC and prediction rate (1 − OER) of 50 models.

Figure 2c is the test AUC value of the models, and Figure 2d is the (1 − OER) + AUC result. The evaluation results showed that the test AUC values of L, LQ models were greater than 0.8, suggesting that the models worked well and had a high prediction accuracy. Additionally, the AUC values of H, LQH, and LQHPF were lower than 0.8. The AUC curve is basically the same as the trend of (1 − OER) + AUC. The evaluation results showed that the (1 − OER) + AUC values of the L and LQ models were almost greater than 1.6. Therefore, when FC is L and LQ, the model is better than H, LQH, and LQHPF. When FC = LQ and RM = 5, the accuracy is the highest, the test AUC is 0.83, and (1 − OER) + AUC is 1.75, which shows that the model has good reliability under this parameter.

### 3.2. Suitable Area Analysis

The predicted distribution of insect pest is consistent with the PWD's current distribution. The Maxent model was established with the 11 variables, according to the ranking results of the influence factors. The study areas were divided into none, low, medium, and high—4 types of suitable areas—by artificial classification (Figure 3). The result was masked by sub-compartments. The distribution of different types of suitability in Yichang City was divided into three regions. The distribution of high-suitability areas was in a Y-shape, the north and south part of the study area are the none-suitability areas, and the high- and medium-suitability areas are concentrated in the east. The medium-suitability and high-suitability areas have an area of 7214.34 km$^2$, accounting for 34.5% of the city. Among them, the high-suitability area covers an area of 3763.43 km$^2$, accounting for 17.9% of the city, it is mainly distributed in Dangyang City, Yiling District, Municipal District, Yidu District, and Yuan'an County. Medium-suitability areas are distributed in the east of Yichang, located in Dangyang City, Zhijiang County, Yuan'an County, and Xiling, Wujiagang of Municipal District. The percentage of medium-suitability areas is 16.5%. The percentage of none-suitability areas is the highest proportion, reaching 50.1%, and are widely distributed in the west of Yichang City, located in the Wufeng City, Dangyang City, Changyang City, Zhijiang City, and Xingshan City.

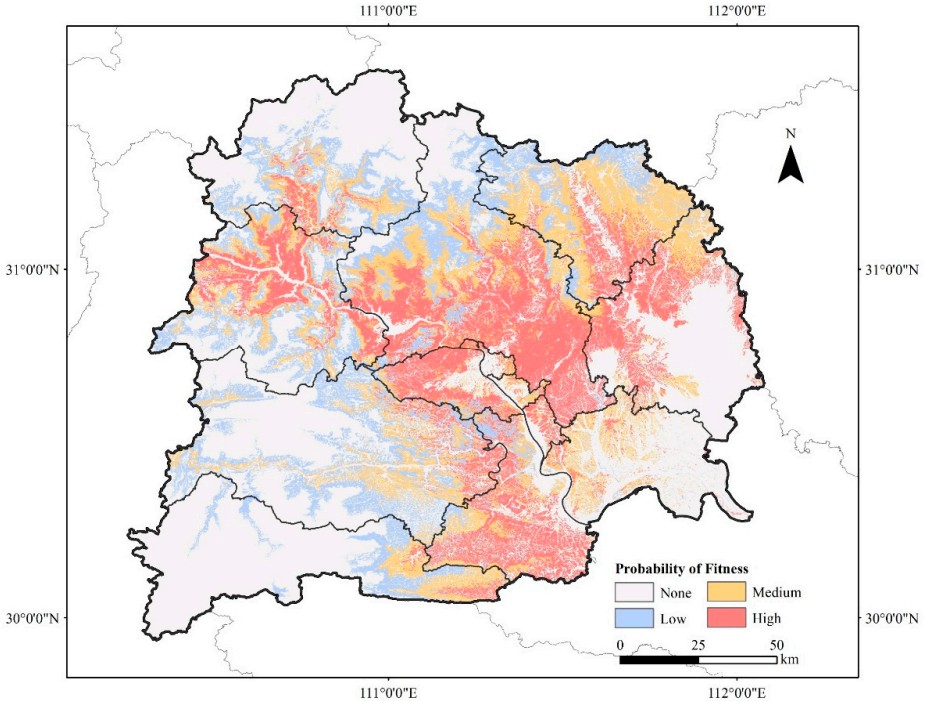

**Figure 3.** Prediction of potential suitable area for PWD in Yichang City.

The jackknife test was used to generate a response map of factors in the model (Figure 4). Maximum temperature, solar radiation, and shape showed a positive cor-

relation with the spread of PWD. Elevation, distance to road, distance to river, soil moisture, precipitation, and cohesion all have a negative correlation to the spread of PWD. Additionally, the categorical data, such as age of stand and landform, both reflect strong differences in different categories. When the elevation increases, the existence probability of PWD begins to decrease and eventually approaches zero. Both precipitation and soil moisture showed the relationship between the suitable probability of PWN and humidity, the response curves of the two variables indicate that drought stress promotes the occurrence of PWD. Temperature is one of the core factors of PWD, has an important influence on the spread of PWD. When the temperature is between 20.5 and 32.2, the suitable probability of PWN increases with the increase in temperature. The age of forest is an extremely important factor in the prediction of PWN suitable areas. In the forest regions, near-mature forest has the highest probability for PWN, half-mature forest has the secondary probability of PWN, and the young, mature, and overmature regions have lower suitable probability. Additionally, the response curve showed that class 11, 21, and 31 had a higher probability for PWD.

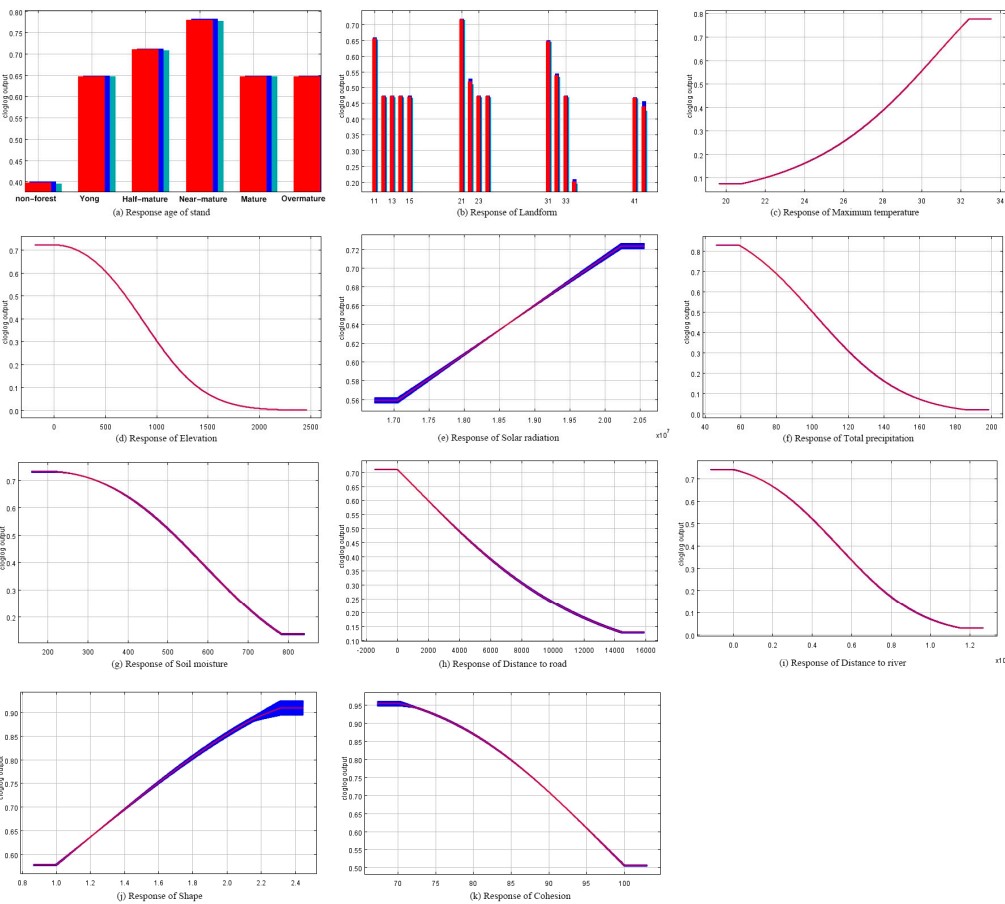

**Figure 4.** Response curve characterizing how each variable affected the Maxent predictions. The curves show the mean response of the 5 replicate Maxent runs (red) and the mean +/− one standard deviation (blue, two shades for categorical variables). (ALOS-derived landform classes: 11—peak/ridge (warm); 12—peak/ridge; 13—peak/ridge (cool); 14—mountain/divide; 15—cliff; 21—upper slope (warm); 22—upper slope; 23—upper slope (cool); 24— upper slope (flat); 31—lower slope (warm); 32—lower slope; 33—lower slope (cool); 34—lower slope (flat); 41—valley; 42—valley (narrow).)

### 3.3. Group Predictive Analysis of Influence Variables

A prediction plan was designed to explore the variable combination for predicting the distribution of PWN. Some variables have low permutation importance in the model.

The factors are divided into three categories: natural factors, landscape factors, and human factors. The variables were divided into three groups, as follows: non-landscape factors, non-human factors, and natural factors. Additionally, we masked the result by sub-compartment. The results were compared with full-factor prediction. The results showed that the distribution of existence suitability is basically similar under the prediction of the three groups (Figure 5).

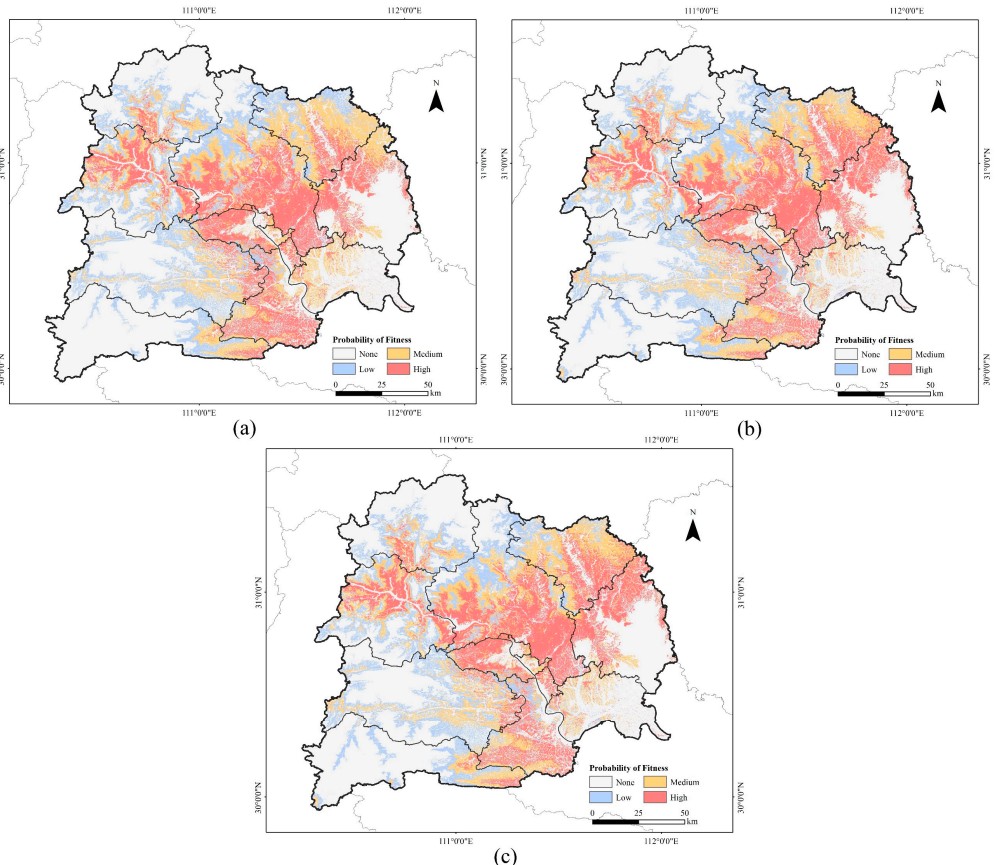

**Figure 5.** Prediction of potential suitable area for PWD in Yichang City in three groups. (**a**) The prediction result of Maxent model with non-landscape factors; (**b**) the prediction result of Maxent model with non-human factors; (**c**) the prediction result of Maxent model with nature factors.

The distribution of suitable areas of various types of suitability was not significantly different from the full-factor prediction results. Landscape factors did not show a significant effect. The prediction result of the non-landscape group shows that the percentage of high-suitability areas is 18.2%, increased by 0.3%, and the percentage of medium-suitability areas were not significantly different from the full-factor prediction. The non-human group and the natural factor group have a similar distribution of existence suitability, and they are different from the existence suitability distribution of the full-factor prediction. The risk of PWD's occurrence was high in Dangyang and Yuan'an, and the medium-suitability areas and high-suitability areas increased. For the whole study area, the percentage of the none-suitability region decreased by 0.6% and 0.8% according to the prediction results of non-human group and natural group. Additionally, the percentage of high-suitability areas increased by 0.7% and 1%.

### 3.4. Variables of Regional Difference

The permutation importance of factors is different due to differences in climate and other conditions in different regions. There are various factors affecting the distribution of PWD. The change of the permutation importance of factors in 10 regions of Yichang City

was analyzed by the jackknife method (Figure 6). Additionally, several regions are merged as a municipal district (i.e., Dianjun, Wujiagang, Xiling, Xiaoting). As we mentioned before, permutation importance reflects the importance of that factor to the predicted model. Higher values mean important factors, and lower values mean unimportant factors. Altitude and temperature have the highest gains in most areas, including Changyang, Wufeng, Xingshan, Yuanan, and Zigui. The correlation between precipitation and soil moisture is relatively high, and these two humidity variables show high gains in Dangyang, Wufeng, Yidu, Yuanan, Zhijiang, and Zigui. In addition, the solar radiation has a significantly high gain in Zhijiang and Zigui. Human factors have important gains in many areas. The distance to road plays an important role in the spread of PWD in Wufeng, Xingshan, and Zigui. While the distance to the Yangtze River provides a significant gain in Wufeng, Yidu, and Zigui. The landscape factors do not play a decisive role in the prediction in all regions, but the cohesion has obvious gains in the municipal district, Dangyang, and Zhijiang.

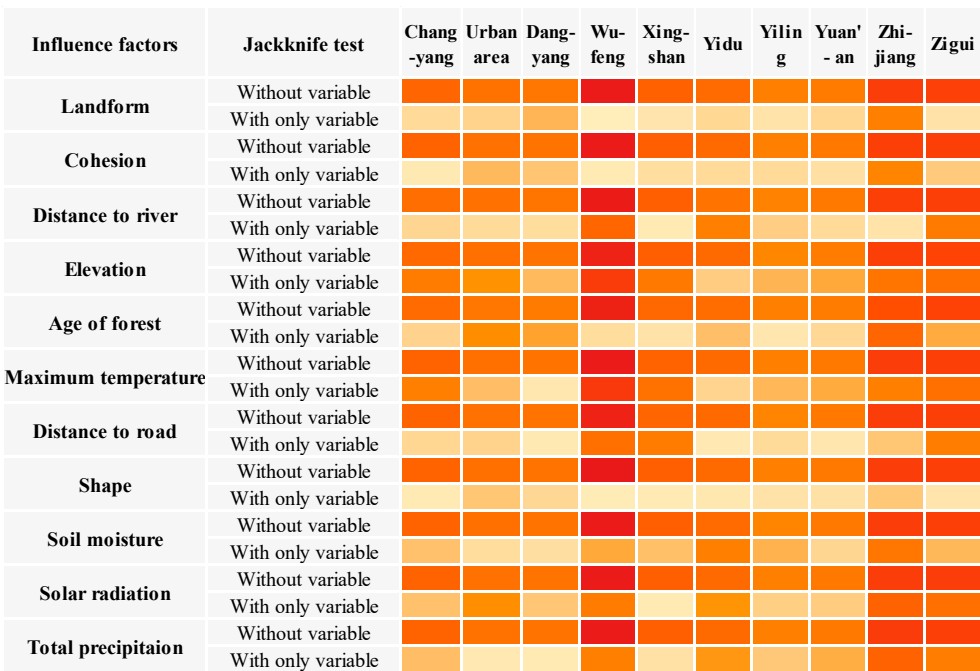

**Figure 6.** Jackknife result of different regions. The picture shows the results of the jackknife test of variable importance in ten regions of Yichang City. The graph is represented by a color scale, where the darker the color, the greater the gain.

## 4. Discussion

The distribution of different types of existence suitability in Yichang City is mainly divided into three parts. The regions of middle-suitability areas and high-suitability areas are distributed in a Y-shape, the none-suitability areas are concentrated in the north and the south parts of the city, and the low-suitability areas are concentrated in the eastern city. The distribution of the high-suitability areas and the none-suitability areas are similar to the distributions of the temperature and elevation, and the high-suitability areas are concentrated in the areas with high temperature, low elevations, and intensive precipitation. The high-suitability areas are mainly distributed in Dangyang, Yiling and Yuan'an, and the medium-suitability areas are distributed in Yiling, Yidu, Yuan'an and Zigui. The PWD was mainly concentrated in the areas which have convenient transportation, developed economy, and relatively frequent human activities in the early stages. Dead trees appeared in Yiling in 2000, and it was confirmed as PWD in 2006. The extant research showed that the epidemic of Yichang was introduced by the construction of cable reels and other projects from the coastal area [28]. Yiling has well-developed transportation and economy and a

dense network of rivers. Additionally, the Gezhouba Water Control Project and the Three Gorges Dam are both located in Yiling. The frequent human disturbance (i.e., material transportation, personnel exchanges) has a great impact on the forest landscape pattern [30]. Then, the epidemic spread from points to areas by highways and large engineering facilities. The spread of PWD showed a leaping pattern in large scope and long distance. Yuan'an first discovered the PWD in 2012 and became an epidemic area in 2019. The disaster of Dangyang has increased rapidly since 2015, and Dangyang was confirmed as an epidemic area in 2017. The number of dead trees in six towns was severe (i.e., Fenxiang and Letianxi of Yiling, Yuquan Office and Wangdian of Dangyang, Songmuping, and Zhicheng of Yidu City, etc.) and the number of dead trees accounted for 44.25% of the dead trees in the city.

Natural factors still play the most important role in the distribution of the disease, and the climate factors have a great contribution. The development and reproduction of PWN are affected by temperature and humidity, the hot and dry summer can accelerate the development of symptoms by weakening photosynthesis and reducing the resistance of host trees [31]. In the culture medium, the inherent natural growth rate of PWNs is proportional to the temperature in the range of 15–30 °C, and the time required to reach the saturation density decreases with the increase in temperature [32]. The humidity variables include precipitation and soil moisture effect significantly in the distribution of PWD in this study. When water stress occurs in the leaves, photosynthesis weakens. Subsequently, some unknown resistance mechanisms of host trees to nematode activity were weakened, nematodes reproduced, and leaf water stress increased. With this positive feedback mechanism, the symptoms develop rapidly until the trees die [31]. Some studies have shown the moisture content of wood is important for the survival of PWN, and it was confirmed that the survival and movement of nematode increase with the decrease in moisture content [33], which is consistent with the results of this study.

In addition to the above factors, other natural factors also have great influences. Correlation analysis shows that altitude has a high correlation with temperature and precipitation. The extant studies have presented evidence that terrain variables can replace climate variables as influence factors for the distribution of PWD [34]. However, terrain variables can be only used as substitute factors for climate, species, and other factors with biological significance [21]. Altitude has the highest contribution of all the influence factors, and the distribution of suitable probability changes with altitude, and the warm slope had higher existence suitability for PWD. We also discovered the effect of forest age on the distribution of PWD. The existed papers have proved that PWN are more susceptible to infect Masson pine trees that are 5–40 years old, and mainly damage Masson pine trees that are 5–20 years old [35]. The results of this study show that half-mature and near-mature pine forests (20–40 years old) are more likely to be damaged.

Climate variables are not the only determinant of the suitability of PWD—natural enemies, soil constraints, and human activities will also affect the species distribution [2,36,37]. The actual habitat of PWN is smaller than that indicated by the existing models. Landscape factors and human factors were used in this study. The result showed that natural factors still play a decisive role in the existence suitability. However, the distance to the Yangtze River and the distance to road also have great influences on the distribution of PWD. Human activities as the main mode of transmission directly affect the spread speed and scale of the disease. The invasion of non-infected areas is mainly related to wood product flow [38]. Unprocessed wood, especially logs, has been identified as one of the high-risk paths for the spread of PWD [39]. The human activities mainly include the exchange of materials through import and export trade along railways and highways. The important ports are distributed along the Yangtze River, located in Xingshan, Zigui, municipal district, Yidu, Zhijiang, and other places. The prediction results of Yidu District show that the distance to the Yangtze River has the largest gain. Landscape factors worked in some regions. Cohesion has a negative correlation with the spread of PWD. Forest landscapes with high aggregation and good cohesion are relatively difficult to infect with PWD [24]. The shape has a positive correlation with the occurrence of epidemic, which was consistent with Hong's result [25].

The core variables that affect the spatial distribution of PWD are different in different regions. The complexity of Maxent model has a significant impact on its transfer ability. This model simulates the distribution of species between the potential distribution and the actual distribution [40]; however, the model needs to be adjusted case by case, and satisfies the needs of different requirements. The model will be overfitting when transferring to different regions, its predictive power may be reduced [41,42]. The eastern part of the study area is a high-altitude area and has great difference in distribution. The elevation is the most important factor affecting the distribution of PWD. The medium-suitability and high-suitability areas are mainly distributed in the low-altitude area below 700 m. This discovery is consistent with the theory of Ning's research [6,38]. The distribution of precipitation is similar to the altitude. The east part of regions has more precipitation and small difference in precipitation. The western region has little rainfall, and its difference is significant. Therefore, the gains of precipitation in the municipal districts, Dangyang, and Yuan'an areas are very low. Though the precipitation in other eastern regions has a key effect, the gains are less than those of western regions.

## 5. Conclusions

In this research, we conducted a multi-angle, fine-scale study on the risk distribution of PWD. Natural factors, landscape factors, and human factors were selected as influence factors to establish a Maxent model, with which, we explored the distribution of existence suitability, and revealed the response trend of different influence factors. Additionally, this paper explored the variable combination for predicting the distribution of PWN by three prediction groups and implemented a regional difference analysis for the change of the influence factors response trend. The result showed the regions of middle-suitability and high-suitability areas are distributed in a Y-shape, and divided the study area into three parts. The none-suitability areas were concentrated in the north and the south parts of the city, and the low-suitability areas are concentrated in the eastern city. The distributions of the high-suitability areas and the none-suitability areas are similar to the distributions of the temperature and altitude, and the high-suitability areas are concentrated in the region with high temperature, low elevation, and intensive precipitation. Natural factors still play the most important role in the distribution of the disease, but they are not the only determinants of the suitability of PWD. The permutation importance of research variables changes under different climatic environments, transportation conditions, and landscape patterns.

**Author Contributions:** Conceptualization, Z.H. and H.Y.; methodology, Z.H. and H.Y.; validation, Z.H.; formal analysis, Z.H.; investigation, Z.H. and H.Y.; data curation, Z.H., B.Z., G.F. and X.L.; writing—original draft preparation, Z.H.; writing—review and editing, Z.H., B.Z. and H.Y.; supervision, W.H.; funding acquisition, W.H. and H.Y. All authors have read and agreed to the published version of the manuscript.

**Funding:** This research was funded by Strategic Priority Research Program of the Chinese Academy of Sciences (XDA19080304), Major Emergency Science and Technology Project of National Forestry and Grassland Administration (ZD202001), Youth Innovation Promotion Association CAS (2021119), and Future Star Talent Program of Aerospace Information Research Institute, Chinese Academy of Sciences (2020KTYWLZX08).

**Institutional Review Board Statement:** Not applicable.

**Informed Consent Statement:** Not applicable.

**Data Availability Statement:** Data sharing is not applicable to this article.

**Conflicts of Interest:** The authors declare that they have no known competing financial interests or personal relationships that could have appeared to influence the work reported in this paper.

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
