# Peer review of "Risk Prediction and Variable Analysis of Pine Wilt Disease by a Maximum Entropy Model"

_forests, doi:10.3390/f13020342_

Round 1

Reviewer 1 Report

Risk prediction and variable analysis of pine wilt disease by maximum entropy model

Comments
In this ms., the authors work on risk prediction of the pine wilt disease PWD and analyze variables that are responsible for PWD using the Maxent model. PWD has been a serious threat to pine trees. And the Maxent model has been widely used to assess potential risk of a target phenomenon (PWD in this ms.). I think the authors' approach is sound and the results are worth to be published.

Overall, the ms. is well written but I found it a bit lengthy. The authors work on many factors/variables that are related to PWD occurrence and it might be difficult to shorten the ms, further. But if possible, I recommend the authors to slim the ms.

My concern is that some terms used in the Maxent model might be unfamiliar with readers of Forests. I myself am not specialist of entropy models and it was difficult for me to follow the method and some results explained in the ms.

Specifically, the following points are not clear to me.

Line 171: ... and 2 FC(i.e, L, LQ, LQH, H, LQHQT) are used in this study.
L refers to Linear, Q to Quadratic, H to Hinge as explained in line 170-171. And LQ and LQH refers to combination of Linear and Quadratic, combination of Linear, Quadratic and Hinge, I guess. But what does LQHQT refer to?

Line 192: It is not clear what PD, SHDI, Shape refer to. These terms might be self-evident for readers who are familiar with the Maxent model. But at least I didn't understand what these terms mean.

Line 205-209: It was not clear to me how Figure 2.a and 2.b could be interpreted as mentioned in the paragraph (line 205-216).

Line 220- 223: The term LQHPF is not clear to me. What is this?

Line 304: The permutation importance of factor is different due to ...
It is not clear what "permutation importance" means. I guess this term is also often used in the Maxent model. But I recommend the authors to give biological interpretation of "permutation importance".

I recommend the authors to give additional information about these terms for readers not familiar with the Maxent model.

Finally, the ms. is awkward in English in some sentences. I recommend the authors to thoroughly check English.

End of comments

Reviewer 2 Report

  1. General Comments

The paper presented has a very good structure and is very easy to read and to understand. The model presented is a theorical one, presents distinct scenarios of the PWN dissemination well supported by studies already done, contemplating the human activity and landscape patterns.  So, it is an excellent exercise well supported in mathematical models.  

  1. Section by section

2.1. Introduction:

Introduction is very easy to read, very comprehensible and has a lot of references to consolidate the affirmations made.

2.2. Material and Methods:

Material and Methods are very clear and allow to understand the study. In case of interest or necessity the description of methods used allows to replicate the assay.

2.3. Results:

Results are well presented; graphic component gives reliable information easy to interpretate. 

2.4. Discussion:

Discussion is well conducted and very interesting to read.

  1. Suggestions

Some small mistakes have been detected.

Line 57,  please write countries

Line 115, please write P. elliottii
